# Serum Total Antioxidant Capacity (TAC) and TAC/Lymphocyte Ratio as Promising Predictive Markers in COVID-19

**DOI:** 10.3390/ijms241612935

**Published:** 2023-08-18

**Authors:** Zoltán Horváth-Szalai, Rita Jakabfi-Csepregi, Balázs Szirmay, Dániel Ragán, Gerda Simon, Zoltán Kovács-Ábrahám, Péter Szabó, Dávid Sipos, Ágnes Péterfalvi, Attila Miseta, Csaba Csontos, Tamás Kőszegi, Ildikó Tóth

**Affiliations:** 1Department of Laboratory Medicine, Medical School, University of Pécs, 7624 Pécs, Hungary; ritacsepregi93@gmail.com (R.J.-C.); szirmay.balazs@pte.hu (B.S.); ragandaniel@hotmail.com (D.R.); peterfalvi.agnes@pte.hu (Á.P.); miseta.attila@pte.hu (A.M.); 2János Szentágothai Research Centre, University of Pécs, 7624 Pécs, Hungary; 3Department of Anaesthesiology and Intensive Therapy, Medical School, University of Pécs, 7624 Pécs, Hungary; simongerda26@gmail.com (G.S.); kovacs-abraham.zoltan@pte.hu (Z.K.-Á.); szabo.peter@pte.hu (P.S.); csontos.csaba@pte.hu (C.C.); ildikodrtoth6@gmail.com (I.T.); 41st Department of Medicine, Division of Infectious Diseases, Medical School, University of Pécs, 7624 Pécs, Hungary; sipos.david@pte.hu

**Keywords:** serum total antioxidant capacity (TAC), enhanced chemiluminescence, TAC/lymphocyte ratio, predictive value, COVID-19

## Abstract

SARS-CoV-2 infection might cause a critical disease, and patients’ follow-up is based on multiple parameters. Oxidative stress is one of the key factors in the pathogenesis of COVID-19 suggesting that its level could be a prognostic marker. Therefore, we elucidated the predictive value of the serum non-enzymatic total antioxidant capacity (TAC) and that of the newly introduced TAC/lymphocyte ratio in COVID-19. We included 61 COVID-19 (*n* = 27 ward, *n* = 34 intensive care unit, ICU) patients and 29 controls in our study. Serum TAC on admission was measured by an enhanced chemiluminescence (ECL) microplate assay previously validated by our research group. TAC levels were higher (*p* < 0.01) in ICU (median: 407.88 µmol/L) than in ward patients (315.44 µmol/L) and controls (296.60 µmol/L). Besides the classical parameters, both the TAC/lymphocyte ratio and TAC had significant predictive values regarding the severity (AUC-ROC for the TAC/lymphocyte ratio: 0.811; for TAC: 0.728) and acute kidney injury (AUC-ROC for the TAC/lymphocyte ratio: 0.747; for TAC: 0.733) in COVID-19. Moreover, the TAC/lymphocyte ratio had significant predictive value regarding mortality (AUC-ROC: 0.752). Serum TAC and the TAC/lymphocyte ratio might offer valuable information regarding the severity of COVID-19. TAC measured by our ECL microplate assay serves as a promising marker for the prediction of systemic inflammatory diseases.

## 1. Introduction

In December 2019, a new pandemic (coronavirus disease 2019, COVID-19) arose in Wuhan, China caused by a novel beta coronavirus (severe acute respiratory syndrome coronavirus 2, SARS-CoV-2) with a positive-sense single-stranded RNA genome with frequent complications of severe acute respiratory syndrome [1]. The worldwide estimated number of patients who died from COVID-19 worldwide between 1 January 2020 and 31 December 2021 was around 18,200,000 [2]. The symptoms vary from a mild nose run to severe acute respiratory distress syndrome (ARDS), followed by multiorgan failure leading to death. Patients can be divided into three groups based on symptoms: Non-severe or even asymptomatic, mild to moderately, and critically ill, requiring life-sustaining treatment in an intensive care unit (ICU) [3].

The appropriate diagnosis of SARS-CoV-2 infection was based on reverse transcription real-time polymerase chain reaction (RT-PCR) [4], and clinicians tried to follow the course of the disease by multiple routine laboratory (leukocyte and lymphocyte counts, cytokine levels, serum ferritin, lactate dehydrogenase [LDH] activity, plasma D-dimer) and clinical parameters [5,6]. The most important issue in the management of COVID-19 was to detect when the cytokine storm began because its early recognition and treatment might have led to favorable outcomes.

Looking at the pathophysiology of COVID-19, Domingo et al. defined four feedback loops in the development of the symptoms [7]. Besides the vicious viral loop, the hypercoagulation loop characterized by increased microthrombi formation, and the Angiotensin-converting enzyme 2 (ACE2)/angiotensin (Ang)-loop, there is the hyperinflammatory loop, in which oxidative stress (OS) might be one of the reasons for the development of severe organ failure due to cellular damage. OS refers to the imbalance between reactive oxygen species (ROS) and the antioxidant system. ARDS in severe SARS-CoV-2 infection is suggested to partially rely on the activation of OS chain reactions associated with the innate immune response [8]. The OS level related to the patient sex might be one factor affecting COVID-19 mortality. Due to their higher OS levels, males are more prone to suffering from severe COVID-19 than females [9]. Up to now, previous research groups investigated serum total antioxidant capacity (TAC) in SARS-CoV-2-infected patients with commercially available assays and found differences between patients and controls, however, with discrepant results [10,11,12,13,14,15,16,17,18].

In our work, we quantified the non-enzymatic TAC of the sera of COVID-19 patients and that of healthy controls with an enhanced chemiluminescence (ECL) microplate assay previously validated by our research group [19]. Moreover, we investigated the TAC/lymphocyte ratio as a novel marker. Our further aim was to assess the predictive value of serum TAC and the TAC/lymphocyte ratio regarding the severity of COVID-19.

## 2. Results

### 2.1. Demographic and Clinical Data of the Enrolled Patients

Patients’ demographic data and clinical characteristics are presented in Table 1. The total number of unvaccinated COVID-19 patients was 61 (*n* = 27 ward and *n* = 34, ICU patients), and that of the control group was *n* = 29. ICU patients were significantly older than ward and control patients. There was no difference between ICU and ward patients regarding sex, body mass index (BMI), and hospital treatment length. Based on a literature review, we examined hypertension and diabetes mellitus prevalence as predisposing factors to COVID-19 [20], and our data showed that hypertension and diabetes mellitus were more prevalent in ICU than in ward patients. ICU patients had a significantly higher mortality rate compared to ward patients. There was no difference in the timing of the symptoms’ appearance of COVID-19 at home; however, when compared to ward patients, ICU patients had more severe symptoms on admission based on SOFA, SAPS II scores, significantly higher CT scores, and their Horowitz quotient (PaO_2_/FiO_2_) already showed severe ARDS. AKI was prevalent only in the ICU patient group.

### 2.2. Admission Laboratory Parameters of COVID-19 Patients and Controls

As illustrated in Table 2, absolute leukocyte count was higher in ICU than in ward and control patients (*p* < 0.05). ICU and ward patients exhibited markedly reduced absolute lymphocyte counts when compared with controls (Table 2, Figure 1B) (*p* < 0.001). There was also a significant difference between the two COVID-19 patient groups regarding absolute lymphocyte counts. Markedly increased D-dimer, se-ferritin, hs-CRP levels and LDH activity were present in both COVID-19 patient groups; moreover, D-dimer levels and LDH activity were significantly higher in ICU than in ward patients. IL-6 and hs-TnT levels also increased more in ICU than in ward patients (*p* < 0.001). Decreased albumin levels were present in both COVID-19 patient groups when compared to controls (*p* < 0.001), with the lowest levels in ICU patients. ICU patients had higher se-creatinine levels than controls (*p* < 0.001), and higher se-urea concentrations than ward patients and controls (*p* < 0.001). Decreased total cholesterol and higher triglyceride and uric acid levels were found in COVID-19 patients when compared to controls (*p* < 0.05).

### 2.3. Serum Total Antioxidant Capacity Levels and Total Antioxidant Capacity/Lymphocyte Ratios in COVID-19 Patients and Controls

ICU patients exhibited the highest serum TAC values, while compared to them, lower TAC levels were observed in ward patients (*p* < 0.01) and controls (*p* < 0.001) (Table 2, Figure 1A). Although ward patients tended to have higher TAC levels than controls, this difference was not significant.

We calculated a ratio using admission TAC levels and lymphocyte counts in the hope we might obtain a more robust marker than TAC alone with an additional clinical value. TAC/lymphocyte ratios were the highest in ICU patients (676.50 [300.05–1164.03] µmol/G), and lower TAC/lymphocyte ratios were observed in ward patients (277.79 [224.85–358.61] µmol/G) (*p* < 0.01) and the lowest ratios were found in controls (134.87 [116.64–179.61] µmol/G) (*p* < 0.001) (Figure 1C). In addition, ward patients had higher TAC/lymphocyte ratios than controls (*p* = 0.001).

### 2.4. Predictive Value of Admission Parameters Regarding the Severity of COVID-19

In order to compare the predictive value of the classical admission parameters, TAC, and the TAC/lymphocyte ratio regarding the severity of COVID-19 (whether the patient requires internal medicine or ICU treatment), we performed ROC analysis (Figure 1D). Besides the classical parameters, as LDH (AUC [area under the curve]-ROC value: 0.933; *p* < 0.001), albumin (AUC-ROC: 0.922; *p* < 0.001), SOFA score (AUC-ROC: 0.818; *p* < 0.001), lymphocyte count (AUC-ROC: 0.810; *p* < 0.001), D-dimer (AUC-ROC: 0.768; *p* < 0.01), IL-6 (AUC-ROC: 0.708; *p* < 0.01), and hs-CRP (AUC-ROC: 0.693; *p* < 0.05), the new markers, TAC (AUC-ROC: 0.728; *p* < 0.01) and the TAC/lymphocyte ratio (AUC-ROC: 0.811; *p* < 0.001), also offered significant predictive values. The optimal cut-off value for TAC was found to be 336.11 µmol/L (diagnostic sensitivity: 75%, diagnostic specificity: 62.5%), whereas, for the TAC/lymphocyte ratio, it was 403.68 µmol/G (diagnostic sensitivity: 65.6%, diagnostic specificity: 83.3%).

### 2.5. Diagnostic Capacity of the Studied Parameters Regarding Acute Respiratory Distress Syndrome

On admission, 63.93% (*n* = 39) of the patients suffered from moderate to severe ARDS (Horowitz Index ≤ 200 mmHg), and among them, 32 patients had to be mechanically ventilated and the other 7 were supported by non-invasive oxygen therapy, while 36.07% (*n* = 22) of the patients had mild (*n* = 10) or no (*n* = 12) ARDS, most of them requiring oxygen therapy via nasal cannula. Patients with moderate to severe ARDS had higher CT scores than those with mild or no ARDS (16.92 ± 6.73 vs. 9.77 ± 3.59; *p* < 0.001). Patients with moderate to severe ARDS exhibited a lower lymphocyte count (0.71 [0.44–1.08] vs. 1.02 [0.77–1.66] G/L; *p* = 0.007) than those with mild or no ARDS; however, there was no significant difference between the two groups regarding TAC levels (363.41 [291.59–492.19] vs. 321.53 [265.56–387.44] µmol/L; *p* = 0.065). TAC/lymphocyte ratios were higher (*p* = 0.004) in patients with severe or moderate ARDS (410.08 [295.25–1017.43] µmol/G) than those with mild or no ARDS (285.13 [222.80–402.38 µmol/G] (Figure 2A).

Based on ROC analysis (Figure 2B), from the routine markers, LDH (AUC-ROC: 0.903; *p* < 0.001), albumin (AUC-ROC: 0.823; *p* < 0.001), SOFA score (AUC-ROC: 0.803; *p* < 0.001), D-dimer (AUC-ROC: 0.733; *p* < 0.01), hs-CRP (AUC-ROC: 0.740; *p* < 0.01), lymphocyte count (AUC-ROC: 0.738; *p* < 0.01), ferritin (AUC-ROC: 0.703; *p* < 0.05), and IL-6 (AUC-ROC: 0.724; *p* < 0.01) had diagnostic capacities regarding the differentiation between patients with severe and mild ARDS. The new marker, the TAC/lymphocyte ratio (AUC-ROC: 0.744; *p* < 0.01), also had significant diagnostic value. The optimal discriminatory level for the TAC/lymphocyte ratio was 325.17 µmol/G (diagnostic sensitivity: 72.2%, diagnostic specificity: 60%).

### 2.6. Diagnostic Capacity of the Studied Parameters Regarding Acute Kidney Injury

Of all enrolled COVID-19 patients, 16 (26.23%) suffered from AKI on admission. Among these patients, 37.5% had acute-on-chronic kidney injury, while 62.5% developed AKI without any previously known kidney disease.

Investigating all enrolled COVID-19 patients, those suffering from AKI had higher admission TAC levels (413.06 [361.82–556.36] µmol/L) and lower lymphocyte counts (0.49 [0.39–1.07] G/L) than non-AKI patients (TAC: 326.34 [264.98–388.33] µmol/L, *p* = 0.003; lymphocyte counts: 0.89 [0.69–1.30] G/L, *p* < 0.05). Moreover, AKI patients exhibited higher (*p* = 0.004) TAC/lymphocyte ratios (933.83 [299.99–1495.93] µmol/G) than non-AKI patients (333.19 [231.87–452.84] µmol/G) (Figure 3A). From the classical parameters, se-creatinine (AUC-ROC: 0.975; *p* < 0.001), LDH (AUC-ROC: 0.834; *p* < 0.001), IL-6 (AUC-ROC: 0.777; *p* < 0.01), albumin (AUC-ROC: 0.738; *p* < 0.01), lymphocyte count (AUC-ROC: 0.707; *p* < 0.05), and hs-CRP (AUC-ROC: 0.699; *p* < 0.05) offered diagnostic values regarding AKI (Figure 3B). Moreover, the new markers, TAC/lymphocyte ratio (AUC-ROC: 0.747; *p* < 0.01) and TAC (AUC-ROC: 0.733; *p* < 0.01), also had significant diagnostic capacities regarding AKI. Cut-off values for the diagnosis of AKI regarding the novel markers were the following: For the TAC/lymphocyte ratio, 718.66 µmol/G (diagnostic sensitivity: 66.7%, diagnostic specificity 85.4%); for TAC, 360.29 µmol/L (diagnostic sensitivity: 80%, diagnostic specificity: 58.5%).

### 2.7. Predictive Values of the Investigated Markers Regarding Mortality of COVID-19 Patients

Of the included COVID-19 patients, 26 (76.5%) of ICU patients died in the hospital. Lymphopenia was more expressed in non-survivor (0.57 [0.41–0.79] G/L) than in survivor patients (1.05 [0.78–1.39] G/L) (*p* < 0.001). Moreover, non-survivors had higher TAC/lymphocyte ratios (676.49 [325.24–1227.22] µmol/G) than survivors (292.47 [226.49–397.27] µmol/G) (*p* < 0.001) (Figure 4A).

ROC analysis (Figure 4B) revealed that among the conventional parameters, LDH (AUC-ROC: 0.885; *p* < 0.001), albumin (AUC-ROC: 0.809; *p* < 0.001), lymphocyte count (AUC-ROC: 0.773; *p* < 0.001), SOFA scores (AUC-ROC: 0.723; *p* < 0.01), D-dimer (AUC-ROC: 0.715; *p* < 0.01), IL-6 (AUC-ROC: 0.699; *p* < 0.05), and hs-CRP (AUC-ROC: 0.682; *p* < 0.05) had predictive values regarding the mortality of patients. The new marker, the TAC/lymphocyte ratio (AUC-ROC: 0.752; *p* = 0.001), also offered significant predictive capacity. The cut-off value for the TAC/lymphocyte ratio was 467.09 µmol/G (diagnostic sensitivity: 61.5%, diagnostic specificity: 87.1%).

### 2.8. Spearman’s Correlation Results

The TAC/lymphocyte ratio positively correlated with LDH (Spearman’s rho, ρ = 0.603, *p* < 0.001), urea (ρ = 0.591, *p* < 0.001), hs-CRP (ρ = 0.568, *p* < 0.001), TAC (ρ = 0.544, *p* < 0.001), triglyceride (ρ = 0.483, *p* < 0.001), creatinine (ρ = 0.433, *p* < 0.001), uric acid (ρ = 0.310, *p* < 0.05), and D-dimer (ρ = 0.308, *p* < 0.05). There was an inverse correlation between the TAC/lymphocyte ratio and lymphocytes (ρ = −0.924, *p* < 0.001), albumin (ρ = −0.612, *p* < 0.001), and total cholesterol (ρ = −0.504, *p* < 0.001). From the clinical parameters, the TAC/lymphocyte ratio positively correlated with SOFA (ρ = 0.465, *p* < 0.001), SAPS II (ρ = 0.491, *p* < 0.001), CT scores (ρ = 0.338, *p* < 0.01), fraction of inspired oxygen (FiO2; ρ = 0.522, *p* < 0.001), and ICU treatment days (ρ = 0.462, *p* < 0.001).

TAC positively correlated with uric acid (ρ = 0.609, *p* < 0.001), urea (ρ = 0.523, *p* < 0.001), triglyceride (ρ = 0.449, *p* < 0.001), creatinine (ρ = 0.439, *p* < 0.001), D-dimer (ρ = 0.380, *p* < 0.01), hs-TnT (ρ = 0.331, *p* < 0.05), and LDH (ρ = 0.309, *p* < 0.01). TAC inversely correlated with albumin (ρ = −0.273, *p* < 0.05). From the clinical parameters, TAC positively correlated with SOFA (ρ = 0.552, *p* < 0.001), SAPS II scores (ρ = 0.487, *p* < 0.001), FiO2 (ρ = 0.323, *p* < 0.05), and ICU treatment days (ρ = 0.323, *p* < 0.05).

## 3. Discussion

Numerous studies suggest that OS plays an important role in the pathogenesis of SARS-CoV-2-induced COVID-19. It could be a significant factor for effective virus replication and might contribute to subsequent virus-induced life-threatening complications [21,22]. In our study, we indirectly assessed the degree of OS in the sera of COVID-19 patients and controls by measuring non-enzymatic TAC with a validated ECL-TAC microplate assay. So far, only one study was conducted by our research group where the same ECL-TAC microplate assay was used for the investigation of TAC among septic and control patients [19]. Other investigators also measured serum TAC in COVID-19 patients [10,11,12,13,14,15,16,17,18], but they used commercially available assays, e.g., colorimetric assays utilizing FRAP (Ferric Reducing Antioxidant Power) or ABTS (2,2-azino-bis(3-ethylbenzothiazoline-6-sulfonic acid) as substrates.

Our demographic findings are similar to those of previous studies where severely ill patients were older and had comorbidities [23,24]. Similar to previous studies, our ICU patients had higher admission clinical severity and CT scores than ward patients. In addition, both ICU and ward patients had pronounced lymphopenia, increased D-dimer, ferritin, hs-CRP, IL-6 levels, and LDH activity [25,26]. In accordance with other studies, LDH had predictive value regarding disease severity, organ failure, and mortality of COVID-19 patients [27,28]. LDH is a marker for tissue damage, and based on its positive correlation with TAC, we suggest that TAC also refers to the degree of tissue injury.

We noted decreased se-total cholesterol and increased triglyceride levels in COVID-19 patients, which are in accordance with previous findings [29], where low cholesterol and high triglyceride concentrations were measured during the hospitalization of severe COVID-19 patients. As in other acute infections, the increase in triglyceride levels might be a result of increased hepatic VLDL production and decreased removal of triglyceride-rich lipoproteins. Along with lipid profile changes, hypoalbuminemia was also found in our patients. It corresponds to the results of other studies where low albumin levels were associated with poor prognosis [30]. Hypoalbuminemia can be explained by possible capillary leak syndrome and decreased hepatic synthesis [31]. In addition, during the infection-caused OS, an excessive amount of dysfunctional albumin is generated and degraded rapidly. Complementary to our findings, Huyut et al. [32] estimated antioxidant-oxidant levels in COVID-19 patients using laboratory parameters with expert models and found that lymphocyte count, ferritin, D-dimer, leukocyte count, and hs-CRP were reliable parameters for that purpose. Ferritin did not prove to be a significant predictive marker in our study; nevertheless, many research groups identified hyperferritinaemia as a red flag of systemic inflammation [33].

The findings of Karkhanei et al. [14] in echo with ours indicated that, although TAC levels did not differ between ward and control patients, ICU patients had significantly higher TAC levels than controls. The observations of Golabi et al. [13] are also similar to ours, where COVID-19 patients had higher TAC levels than controls after the administration of vitamin D supplementation, although our patients did not receive vitamin D at the time of blood sampling. In contrast to our observation, in the study carried out by Martín-Fernández et al. [15], higher activity of antioxidant enzymes but lower TAC (ABTS and FRAP) levels were noted in COVID-19 patients than in controls. Similar to their observation, Doğan et al. [12] indicated decreased total antioxidant status in all SARS-CoV-2 infected patients when compared to controls. Yaghoubi et al. [18] also found lower TAC levels in COVID-19 patients when compared to healthy individuals and patients. These discrepancies regarding serum TAC levels in COVID-19 patients vs. controls between different studies could be attributed to the different antioxidant assays applied. The indicators, oxidants, standards, and principles of the TAC measurement can significantly differ from each other. Generally, in the human serum, non-enzymatic antioxidant compounds include low-molecular-weight molecules, such as uric acid, protein thiol-groups, bilirubin, and vitamins (e.g., C and E). The contribution of various antioxidants to the non-enzymatic antioxidant capacity of serum varies depending on the assay used. Similar to our recent findings, in our previous study carried out among septic patients [19], TAC strongly positively correlated with uric acid. Based on our results, it seems that the nitrogenous waste uric acid contributes to a significant portion of TAC measured by our ECL microplate assay.

We introduced a new marker, the TAC/lymphocyte ratio, which has not been investigated previously. Lymphopenia is a well-known hematological finding among SARS-CoV-2 infected patients [5], and because it changes inversely to TAC, we suggested that the ratio of them might provide a robust marker with additional predictive value. Regarding COVID-19 severity, based on ROC analyses, we found that TAC/lymphocyte ratios and TAC had significant discriminatory values besides the classical parameters. Similar to us, Doğan et al. [12] and Karkhanei et al. [14] also noted a discriminatory role of total antioxidant status regarding the diagnosis of COVID-19. We also investigated the diagnostic/predictive ability of our new markers in organ failures. In the diagnosis of severe ARDS, the TAC/lymphocyte ratio showed additional diagnostic value. Moreover, both TAC and the TAC/lymphocyte ratio offered diagnostic values regarding AKI. As far as we know, no previous studies have elucidated the predictive capacity of TAC and the TAC/lymphocyte ratio in various organ failures among COVID-19 patients. Recently, it has been ascertained that the overwhelming production of ROS in COVID-19 patients results in tissue damage leading to disease progression. Due to enhanced OS, an excessive amount of the neutrophil extracellular trap is formed, which is suggested to be involved in immunothrombosis and organ damage [34]. Because of the observed positive correlation between uric acid and TAC, to the analogy of the TAC/lymphocyte ratio, we also formed the uric acid/lymphocyte ratio and investigated its diagnostic/predictive values regarding the four outcomes (severity of COVID-19, ARDS, AKI, and mortality) of our study, which was comparable to that of the TAC/lymphocyte ratio. Our observation seems to prove our suggestion, namely, that uric acid contributes to a significant portion of the TAC measured by our ECL-TAC assay. However, since TAC could also derive from other low-molecular-weight compounds apart from uric acid, it might provide more complex information regarding the oxidant/antioxidant status of the patients than uric acid alone. Whether the uric acid/lymphocyte ratio could be used interchangeably with the TAC/lymphocyte ratio for the assessment of systemic viral diseases requires further investigation.

We also observed that the TAC/lymphocyte ratio offered predictive value regarding COVID-19 mortality besides the classical markers. Karkhanei et al. [14] obtained similar results to ours regarding TAC and COVID-19 mortality.

For assessing the prognosis of COVID-19, several markers have been investigated. The unbalanced immune response due to severe SARS-CoV-2 infection was mainly demonstrated by an increase in neutrophil granulocyte counts and a reduction in lymphocyte counts, thus altering the neutrophil-to-lymphocyte ratio (N/L ratio), which showed an association with COVID-19 severity and mortality [35,36]. Because not every patient had a full complete blood count in our study, the investigation of the N/L ratio was not feasible. As also presented in our results, serum lactate dehydrogenase (LDH) and albumin are also prognostic markers in COVID-19. In addition, the combinations of clinical and laboratory parameters were also proven to be predictive markers, e.g., classical severity scores (SOFA, SAPS-II) and CURB-65 (based on the following: Confusion, uremia, respiratory rate, blood pressure, age ≥ 65-years-old), but also the new MuLBSTA score (based on the following parameters: Multilobar infiltration, Hypo-lymphocytosis, Bacterial coinfection, Smoking history, hypertension, and age) [37,38]. However, there is no specific laboratory parameter to verify COVID-19 cases except the real-time RT-PCR method to directly identify the presence of the virus in the patients.

Our study bears some limitations. The number of unvaccinated ICU and ward patients included in the study is limited, as we started our study parallel to the introduction of vaccination. Besides TAC, it would have been interesting to investigate antioxidant enzymes and direct OS markers; in addition, further follow-up of the patients might also offer intriguing additional clinical information. The results presented here would require validation in larger clinical studies.

## 4. Materials and Methods

### 4.1. Type of Study

This monocentric, prospective, observational cross-sectional study was conducted over the time period of 30 April 2021 to 28 February 2022, during the third wave of the COVID-19 pandemic, characterized by delta variant viral infection. Blood sampling from unvaccinated COVID-19 patients was performed at the Department of Anaesthesiology and Intensive Therapy and at the 1st Department of Medicine, Division of Infectious Diseases of the University of Pécs, Medical School, Hungary. We recruited healthy unvaccinated subjects without any SARS-CoV-2 infection and formed a control group. Sampling from controls was carried out at the Department of Laboratory Medicine, University of Pécs, Medical School, Hungary.

### 4.2. Patients

The target groups of the study were unvaccinated and SARS-CoV-2-infected patients admitted to the ICU or the Division of Infectious Diseases. Written informed consent was obtained from all patients or from their surrogates.

The inclusion criteria for COVID-19 patients were as follows:Over 18 years of age.Proven SARS-CoV-2 infection (positive RT-PCR result from nasopharyngeal sample).Admission to the hospital primarily because of COVID-19.One or no organ failure in the case of mild symptoms (ward patients) and a minimum of two organ failures in case of moderate or severe symptoms (ICU patients).Increased levels of at least two of the following laboratory parameters: LDH, ferritin, D-dimer, IL-6, and high-sensitivity C-reactive protein (hs-CRP) [39].

The exclusion criteria for COVID-19 patients were as follows:Vaccination against COVID-19.Presence of any autoimmune disease.Presence of microbiologically proven bacterial infection.Second or more hospital admission due to COVID-19.Therapy (antiviral therapy, steroids, vitamins) for COVID-19 prior to ICU or ward admission.Pregnancy.Withdrawal of consent.

The inclusion criteria in the case of controls were as follows:Apparently healthy volunteers with no medication, no symptoms of any current infection, physiological complete blood count, hs-CRP levels below the diagnostic laboratory’s reference limit.The exclusion criteria for control subjects were:Under 18 years of age.Inflammatory or infectious diseases, or any kind of internal medicine disorders.

### 4.3. Measurements

#### 4.3.1. Sample Collection

In the case of COVID-19 patients, sampling was performed right after clinical diagnosis, at hospital admission. A blood sample was drawn via the puncture of a vein or through the arterial catheter into plain blood collection tubes (Greiner, Austria). In the case of controls, a venous blood sample was obtained ambulatory via the antecubital vein into plain and EDTA tubes using a closed blood sampling system (Greiner, Austria). Clotted blood samples were centrifuged for 10 min at 1500× *g* and native sera were stored in Eppendorf tubes at −80 °C until further analyses. Results of routine parameters were collected from the laboratory information system and patients were anonymized. Data were collected for complete blood count, D-dimer, ferritin, hs-CRP, high-sensitivity troponin T (hs-TnT), IL-6, LDH, and kidney and liver function tests. Results of RT-PCR tests detecting SARS-CoV-2 were also collected, which were performed by LightMix^®^ Modular E- and N-gene kits on the Cobas Z 480 PCR platform (Roche Diagnostics GmbH, Mannheim, Germany).

#### 4.3.2. Laboratory Measurements

Serum TAC was measured by luminol-based enhanced chemiluminescence (ECL) antioxidant capacity microplate assay previously validated by our research group [19] at the Szentágothai Research Centre (University of Pécs, Hungary). For the assay, luminol (3-aminophthalhydrazide), 4-iodophenol, Trolox (6-hydroxy-2,5,7,8-tetramethylchroman-2-carboxylic acid), and horseradish peroxidase (POD) were obtained from Merck Life Science Ltd., Darmstadt, Germany, 10 M H_2_O_2_ was purchased from Molar Chemicals, Budapest, Hungary. As a standard for each measurement, Trolox was used (0–100 μmol/L). For the analysis, white 96-well microplates (Per-Form Hungaria Ltd., Budapest, Hungary) and a Biotek Synergy HT plate reader (Agilent, Santa Clara, CA, USA) were applied. Furthermore, a 20 μL blank/standard/sample was pipetted into the microplate wells in triplicates. After that, 270 μL of the POD-ECL solution was pipetted and the plate was shaken for 10 s. The ECL reaction was triggered by the automatic injection of 20 μL of 1.5 mmol/L H_2_O_2_ solution. Monitoring of the luminescent reaction in the samples was performed for 10 min at 64 s measuring intervals. TAC was assessed with the help of the total light output (area under curve, AUC) of the corresponding standards. TAC of the samples was calculated using the equation of the standard curve and was expressed as Trolox equivalent (TE) in μmol/L. Sera from patients were 5–7.5-fold diluted depending on the initially obtained TE concentration. In the case of diluted sera, TE was multiplied by the applied dilution factor. An average TAC was calculated in the case of every sample based on 3 independent measurements.

Routinely measured parameters were assessed by automated analyzers in a fully accredited routine laboratory at the Department of Laboratory Medicine (University of Pécs, Hungary; accreditation registration number: NAH-9-0008/2021).

#### 4.3.3. Collection and Assessment of Clinical Parameters

Collected clinical data included the basic demographic data of the patients (age, sex, body mass index [BMI], and comorbidities), initial symptoms, oxygen support therapy on admission and blood gas values, cardiovascular parameters, chest computer tomography (CT) score [40], and clinical severity scores (simplified acute physiology II [SAPS II] [41], sequential organ failure assessment [SOFA] scores [42]). As a follow-up, we recorded hospital mortality. ARDS was assessed based on the Berlin Definition [43], and acute kidney injury (AKI) was defined according to KDIGO AKI Guideline [44].

#### 4.3.4. Statistical Analysis

Data were analyzed with IBM SPSS Statistics for Windows, Version 28 program. The distribution of data was evaluated by the Shapiro–Wilk test. Data with normal distribution are presented as mean ± standard deviation (SD) and analyzed with an independent-sample *t*-test. Non-normally distributed data were expressed as medians and interquartile ranges (IQR). Kruskal–Wallis and Mann–Whitney tests were performed to investigate differences between patient groups. Missing data were excluded from the analysis. Categorical variables were compared by the Chi-squared test followed by Bonferroni post hoc correction. Correlations between quantitative parameters were determined by Spearman’s rank correlation test. Diagnostic and predictive values of the parameters were assessed by receiver operating characteristic (ROC) curves. The significance level was set at *p* < 0.05.

## 5. Conclusions

Based on our results, we suggest that serum TAC and the new marker, the TAC/lymphocyte ratio, could offer valuable information regarding the severity, related organ failures, and outcome of COVID-19. TAC measured by our fast ECL-based TAC microplate assay might be an additional promising marker for the indirect assessment of oxidative stress and the prediction of severe inflammatory diseases.

## Figures and Tables

**Figure 1 ijms-24-12935-f001:**
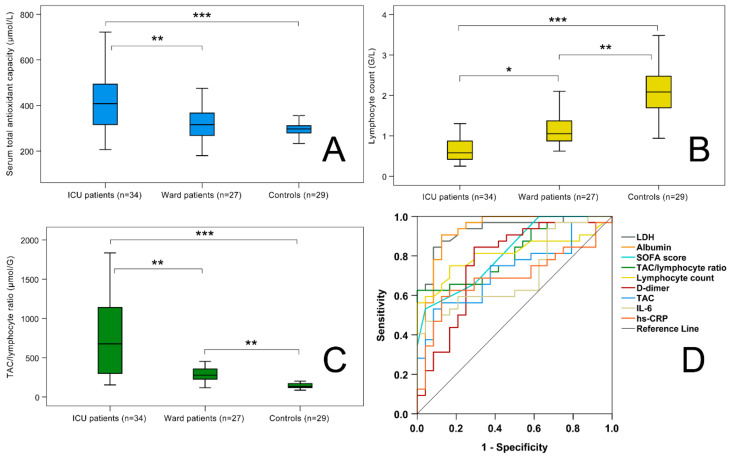
Serum total antioxidant capacity levels (**A**), absolute lymphocyte counts (**B**), and total antioxidant capacity (TAC)/lymphocyte ratios (**C**) in controls, ward, and intensive care unit COVID-19 patients; receiver operating characteristic (ROC) curves of admission laboratory and clinical parameters for differentiating ICU from ward COVID-19 patients (**D**). * *p* < 0.05, ** *p* < 0.01, *** *p* < 0.001. SOFA score: Sequential organ failure assessment score.

**Figure 2 ijms-24-12935-f002:**
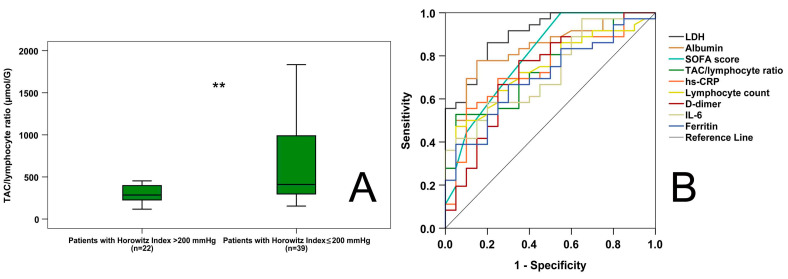
TAC/lymphocyte ratios in COVID-19 patients suffering from mild to no (Horowitz Index > 200 mmHg) and moderate-severe (Horowitz Index ≤ 200 mmHg) acute respiratory distress syndrome (ARDS) (**A**) and diagnostic capacity of the studied markers regarding ARDS investigated by ROC analysis (**B**). ** *p* < 0.01. SOFA score—sequential organ failure assessment score.

**Figure 3 ijms-24-12935-f003:**
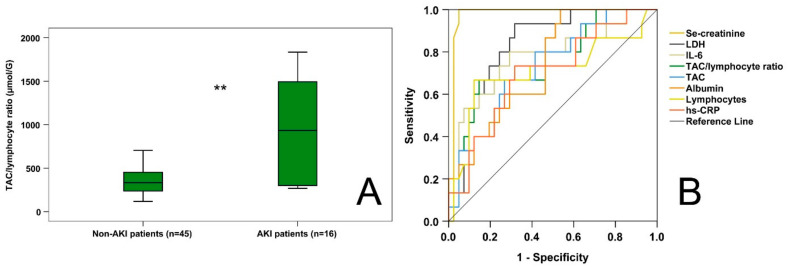
TAC/lymphocyte ratios in COVID-19 patients with and without acute kidney injury (AKI) (**A**); diagnostic capacity of the studied parameters regarding AKI (**B**). ** *p* < 0.01.

**Figure 4 ijms-24-12935-f004:**
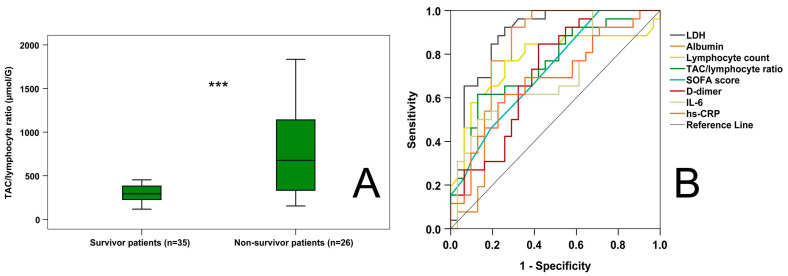
TAC/lymphocyte ratios in survivor vs. non-survivor COVID-19 patients (**A**) and receiver operating characteristic curves of admission laboratory and clinical parameters for predicting hospital mortality in COVID-19 patients (**B**). *** *p* < 0.001. SOFA score—sequential organ failure assessment score.

**Table 1 ijms-24-12935-t001:** Demographic data and SARS-CoV-2 infection characteristics.

		ICU Patients (*n* = 34)	Ward Patients (*n* = 27)	Controls (*n* =29)	Significance
Demography	Age (years)	67 (57–74)	56 (49–61)	43 (36–57)	*p* < 0.01 ^a,b^
	Male (%)	19 (55.9%)	19 (70.4%)	12 (41.4%)	n.s.
	BMI	30.91 (28.0–36.62)	28.60 (26.88–32.69)	-	n.s.
	HT (%)	29 (85%)	10 (37%)	-	*p* < 0.001
	DM (%)	15 (44%)	3 (11%)	-	*p* < 0.01
	LOS (days)	10 (5–15)	8 (6–11)	-	n.s.
	Mortality (%)	26 (76.5)	0 (0)	-	*p* < 0.001
COVID-19 at home	Days at home	5 (4–7)	7 (4–10)	-	n.s.
	Pulmonary (%)	23 (68)	23 (85)	-	n.s.
	Gastrointestinal (%)	6 (18)	10 (37)	-	n.s.
	Others (%)	8 (27)	10 (37)	-	n.s.
COVID-19 on admission	SAPS II score	30 (21.75–42.50)	13 (13–18)	-	*p* < 0.001
	SOFA score	4 (2–5.25)	2 (1–3)	-	*p* < 0.001
	CT score	17.19 ± 7.30	10.78 ± 3.69	-	*p* < 0.001
	PaO_2_/FiO_2_ (mmHg)	80.62 ± 56.19	280.79 ± 120.92	-	*p* < 0.001
Clinical parameters	ARDS severe/moderate (%)	32 (94.1)	7 (25.9)	-	*p* < 0.001
	ARDS mild (%)	2 (5.9)	8 (29.6)	-	*p* < 0.001
	AKI (%)	16 (47.1)	0 (0)	-	*p* < 0.001

Categorical variables are presented as numbers and percentages: *n* (%), continuous data are presented as mean ± SD, median and interquartile ranges (25–75%), number and percentages, the significance level is accepted as *p* < 0.05. BMI—body mass index, DM—diabetes mellitus, HT—hypertension, LOS—length of hospital stay, others (fever, malaise, loss of the sense of smell, loss of appetite), SAPS II—simplified acute physiology score II, SOFA—sequential organ failure assessment score, ARDS—acute respiratory distress syndrome, AKI—acute kidney injury. Superscript lowercase letters refer to post hoc analyses: ^a^ ICU COVID-19 patients vs. controls; ^b^ ICU COVID-19 patients vs. ward COVID-19 patients. n.s.: non-significant.

**Table 2 ijms-24-12935-t002:** Admission laboratory parameters of the studied patients.

Parameters	ICU Patients (*n* = 34)	Ward Patients (*n* = 27)	Controls (*n* = 29)	Significance
Leu (G/L)	9.12 (6.71–12.43)	6.25 (5.58–8.50)	6.39 (5.65–7.43)	*p* < 0.05 ^a.c^
Ly (G/L)	0.58 (0.42–0.92)	1.05 (0.86–1.39)	2.09 (1.68–2.50)	*p* < 0.05 ^a,b,c^
Platelets (G/L)	177.50 (144–259)	217 (146.75–290)	256 (210–288)	*p* < 0.05 ^a^
Ferritin (µg/L)	1157 (559–2266.50)	1035 (534–1878)	-	n.s.
D-dimer (µg/L)	1924 (1292–6489)	814 (610–1270)	-	*p* < 0.001
INR	1.08 (1.03–1.24)	1.09 (1.04–1.14)	-	n.s.
IL-6 (pg/mL)	140.75 (44.53–313.28)	54.80 (24.90–88.40)	-	*p* < 0.01
hs-CRP (mg/L)	131.40 (51–188.68)	64 (43.10–102.50)	1.08 (0.62–1.88)	*p* < 0.001 ^a,b^
LDH (U/L)	903 (659.50–1361.50)	419 (321.50–513)	285 (170.25–335.75)	*p* < 0.05 ^a,b,c^
hs-TnT (ng/L)	37.20 (19.16–93.58)	8.42 (4.06–14.59)	-	*p* < 0.001
Total bilirubin (µmol/L)	6.10 (2.90–7.75)	6.20 (4.13–9.43)	7.50 (5.83–11.98)	n.s.
Albumin (g/L)	30.55 (27.80–32.38)	38.80 (35.45–40.33)	48 (46.25–49.50)	*p* < 0.001 ^a,b,c^
Uric acid (µmol/L)	348 (285.75–511.25)	342.50 (257.50–425.50)	259 (224.50–278.50)	*p* < 0.01 ^a,b^
Cholesterol (mmol/L)	3.09 (1.88–3.51)	3.43 (3.16–3.89)	5.11 (4.43–5.80)	*p* < 0.001 ^a,b^
Triglyceride (mmol/L)	1.73 (1.15–2.72)	1.48 (1.28–1.91)	1.02 (0.68–1.30)	*p* < 0.05 ^a,b^
Creatinine (µmol/L)	112 (77.25–263)	85 (75–100)	74.50 (65.25–88.25)	*p* < 0.001 ^a^
Carbamide (mmol/L)	13.66 (8.39–17.97)	5.39 (4.37–6.90)	4.60 (4.03–5.48)	*p* < 0.001 ^a,c^
TAC (µmol/L)	407.88 (310.22–494.08)	315.44 (266.15–368.49)	296.60 (274.96–314.20)	*p* < 0.01 ^a,c^

Data are expressed as medians and interquartile ranges (25%–75%) given in parentheses. hs-CRP—high-sensitivity C-reactive protein; hs-TnT—high-sensitivity Troponin T; ICU—intensive care unit; IL-6—interleukin 6; INR—international normalized ratio; LDH—lactate dehydrogenase; Leu—leukocytes; Ly—lymphocytes; TAC—total antioxidant capacity. Superscript lowercase letters refer to post hoc analyses: ^a^ ICU COVID-19 patients vs. controls; ^b^ Ward COVID-19 patients vs. controls; ^c^ ICU COVID-19 patients vs. ward COVID-19 patients.

## Data Availability

The data presented in this study are available upon request from the corresponding author.

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
