# Peer review of "Serum Total Antioxidant Capacity (TAC) and TAC/Lymphocyte Ratio as Promising Predictive Markers in COVID-19"

_ijms, 2023, doi:10.3390/ijms241612935_

Round 1

Reviewer 1 Report

The problem is current. The proposed marker for the inflammatory process is interesting. Maybe it could also be applied to the proposed marker for other possible infections in the future. It is certain that this should be tested on a larger number of samples.

Questions that the authors could further clarify:

1.     On the basis of laboratory diagnostics, could any other parameter be taken as a marker for overlooking the disease?

2.     Did the literature search lead to similar research and were similar results obtained?

3.     You used relevant methods to determine the TAC. Could the DPPH method be used?

Round 2

Reviewer 2 Report

I will request minor revision. Please see the attached file in which my review comments are described.
